# Exploring the Cognitive Capacities of Japanese Macaques in a Cooperation Game

**DOI:** 10.3390/ani11061497

**Published:** 2021-05-21

**Authors:** Ryan Sigmundson, Mathieu S. Stribos, Roy Hammer, Julia Herzele, Lena S. Pflüger, Jorg J. M. Massen

**Affiliations:** 1Department of Philosophy, University of Vienna, 1010 Vienna, Austria; ryansigmundson@gmail.com; 2Animal Behaviour and Cognition, Department of Biology, Utrecht University, 3584 CH Utrecht, The Netherlands; m.s.stribos@gmail.com (M.S.S.); roy.hammer@live.nl (R.H.); 3Austrian Research Center for Primatology, 9570 Ossiach, Austria; herzele.julia@gmail.com (J.H.); lena.pflueger@univie.ac.at (L.S.P.); 4Department of Behavioral and Cognitive Biology, University of Vienna, 1090 Vienna, Austria

**Keywords:** inequity aversion, ecological validity, field experiments, loose-string paradigm, partner choice, animal cognition

## Abstract

**Simple Summary:**

Experiments using animal models are often conducted to explore the cognitive capacities of different species and to shed light upon the evolution of behavior and the mind that shapes it. Investigating the cognitions and motivations involved in cooperation is one such area that has attracted attention in recent years. As experiments examining these abilities in natural settings are underrepresented in the literature, our study was conducted in a setting closely resembling the natural environment of the study species so as to retain the social factors that help shape these behaviors. In our experiments, Japanese macaques needed to work together to simultaneously pull two loops in order to release food rewards onto a central platform. Over the course of the experiment, the macaques in our study came to make fewer attempts at the cooperative task when no potential partner was present. Furthermore, following an unequal division of the rewards, macaques receiving lesser rewards were more likely to express stress-related and aggressive behavior. Together, these results suggest that the Japanese macaques in our study understood the importance of having a partner in the cooperative task, paid attention to the relative value of the reward they received from the task and became distressed if their reward was inferior to that of another.

**Abstract:**

Cooperation occurs amongst individuals embedded in a social environment. Consequently, cooperative interactions involve a variety of persistent social influences such as the dynamics of partner choice and reward division. To test for the effects of such dynamics, we conducted cooperation experiments in a captive population of Japanese macaques (*Macaca fuscata*, N = 164) using a modified version of the loose-string paradigm in an open-experiment design. We show that in addition to becoming more proficient cooperators over the course of the experiments, some of the macaques showed sensitivity to the presence of potential partners and adjusted their behavior accordingly. Furthermore, following an unequal reward division, individuals receiving a lesser reward were more likely to display aggressive and stress-related behaviors. Our experiments demonstrate that Japanese macaques have some understanding of the contingencies involved in cooperation as well as a sensitivity to the subsequent reward division suggestive of an aversion to inequity.

## 1. Introduction

Humans may stand out when it comes to intensive cooperation [1], but we are far from alone in possessing these abilities. Examples of cooperation exist throughout the animal kingdom in behavioral domains as diverse as predator defense [2], group hunting [3], cooperative breeding [4] and coalition formation [5]. The study of animal cooperation has flourished in recent years (for a recent review, see [6]), but many open questions remain regarding the cognitive mechanisms that promote and maintain it. Discussion of the subject has been impeded by the numerous definitions of cooperation existing across the literature, so to avoid confusion, throughout this paper we will use the definition, “all interactions or series of interactions that, as a rule (or ‘on average’), result in a net gain for all participants” [7].

The experimental study of animal cooperation has been beset with uncertainty when it comes to determining what level of task understanding and intentionality is actually possessed by the animals engaging in it [8]. Many cooperative tasks could be completed through sheer coincidence by two animals acting independently, i.e., acting apart together (cf. [7]). Coincidental cooperation aside, successful cooperation could itself occur through merely responding to the presence of a conspecific, through attending to their actions or through a full-on understanding of their intentions. Different cognitions underlie these forms of cooperation, and it is for this reason that extreme care must be taken in designing experiments to differentiate between them. 

The setup most frequently used to investigate cooperation in animals was originally developed by Crawford [9] and later modified into the loose-string paradigm by Hirata [10,11]. In this paradigm, two animals must coordinate their actions to pull two ends of a string simultaneously in order to move food rewards close enough to be accessed. If they mistime their attempt or make an attempt in the absence of a partner, the string comes loose and precludes further attempts. The paradigm has proved effective for testing a range of species such as chimpanzees (*Pan troglodytes*) [12], macaques (*Macaca sylvanus*; *Macaca fuscata*; *Macaca fascicularis*) [13,14,15], marmosets (*Callithrix jacchus*) [16], corvids (*Corvus frugilegus*; *Corvus corax*) [17,18], elephants (*Elephas maximus*) [19], wolves (*Canis lupus*) [20], hyenas (*Crocuta crocuta*) [21], otters (*Pteronura brasiliensis*; *Aonyx cinerea*) [22], parrots (*Nestor notabilis*; *Ara glaucogularis*; *Psittacus erithacus*; *Eupsittula aurea*) [23,24,25,26,27] and dolphins (*Tursiops truncatus*) [28], to name but a few.

Even though the design of the loose-string paradigm makes uncoordinated attempts conspicuous, it is still possible for animals to succeed at the task by virtue of arriving to it simultaneously. To counter this, some studies have taken the design a step further and introduced a delay before releasing a prospective cooperative partner to ensure that active coordination is necessary to complete the task [17,18,19,20,21,22,24,25,26,27,28,29]. Success then necessitates waiting behavior on the part of the initially arriving animal, which at minimum indicates understanding that a partner is required to complete the task.

In addition to task-understanding, another, and more general, pitfall of cooperation is that it is open to exploitation by so-called “free riders”—those who would reap the benefits of cooperative interactions while contributing little effort themselves [30]. For cooperation to have evolved in a species, it needs to consistently provide payoffs for those engaging in it. This requires some form of behavioral strategy to avoid interacting with individuals who reliably exploit others. The aversive reaction to inequity in reward distribution, termed inequity aversion (IA), is theorized to play just such a role and may have evolved alongside cooperation [31,32]. According to this theory, individuals compare their own efforts and benefits to that of their partner and judge whether the outcome is equitable or “fair”. A negative assessment may be expressed through protest, agonistic behavior, unwillingness to continue cooperating or partner switching. These IA-mediated reactions may shield cooperation from free-riders, in which case the mechanism may act as a stabilizer in cooperative interactions [31,32]. IA can be divided into disadvantageous IA, in which the actor is on the detrimental side of the distribution, and advantageous IA, in which the actor is on the beneficial side [32]. As disadvantageous IA is much more common amongst animals [32,33], we use the term IA to refer specifically to this form.

The tendency to cooperate with individuals who are neither kin nor mate has emerged as a strong predictor of the existence of IA in a species [31]. Japanese macaques (*Macaca fuscata*) are known to form coalitions with non-kin for the purpose of social support during conflicts [34], making them excellent subjects to investigate this in. Even though the species does not habitually cooperate in the foraging domain, a study of the closely related long-tailed macaques (*Macaca fascicularis*) demonstrated that IA is not domain specific and thus can be generalized from the social to the foraging domain [35]. It is therefore likely that IA will be present in Japanese macaques and detectable in a food-rewarded task. Moreover, previous work established the capacity to succeed at an experimental cooperative task in Japanese macaques [14]. The same study, however, also found considerable variation in the success rates of different populations at the task that could be linked to differences in social tolerance between those two wild populations [14].

We tested cooperation in a population of Japanese macaques kept under conditions resembling those typical of their natural environment [36] in an enclosure large enough to justify use of the term semi-free (cf. [37,38,39,40,41]), which is how we will refer to their conditions from this point onwards. Our first experiment aimed to demonstrate that this particular population is capable of cooperation, then to further examine this capacity by establishing whether individuals understood that having a partner was necessary for their own success. To accomplish this, we employed a cooperation task that required two individuals to coordinate their pulling of opposing loops in order to release food rewards to a central platform. We predicted that individuals would increasingly come to take into account the presence of a partner when making attempts as the experiment progressed and they had opportunity to learn the contingencies of the task. This would be reflected behaviorally by a change in the proportion of attempts to pull made in the presence of a potential partner as well as by an increase in the frequency of waiting behavior, i.e., not pulling until a potential partner has arrived. 

Our second experiment used the same setup, but with the introduction of an unequal reward condition to investigate the presence of IA, or at least an “emotional” response that may be a precursor of it [6], in the species. Here we predicted that individuals would display a higher rate of stress-related and aggressive behaviors when they had received a comparatively smaller reward than their cooperative partner. 

A strategy of reacting to cumulative outcomes over a period of time rather than each instance of disadvantageous outcome leads to greater tolerance of accidental and unintentional unequal outcomes, thereby protecting potentially valuable cooperative bonds. Only if an individual consistently experiences disadvantageous outcomes over multiple collaborations should it act against it [31]. For this reason, we further predicted that individuals would show more frequent stress-related and aggressive behaviors when confronted with an accumulation of disadvantageous outcomes rather than only one.

## 2. Materials and Methods

### 2.1. Study Subjects and Housing 

Our study was conducted at the Affenberg Landskron (Affenberg Zoobetriebsgesellschaft mbH), in Carinthia, Austria, where a population of Japanese macaques (*Macaca fuscata*) resides under semi-free conditions in a ±40,000 m^2^ enclosure located in a coniferous forest. We classify this population as semi-free because they live in a large enclosure with conditions resembling their natural environment, living space and group size [36], while having the opportunity to naturally form social groups. The population originates from Minoo City, Japan, and was introduced to the area in 1996. At the beginning of data collection for these studies, the enclosure contained 164 individuals living in a single group. The population consisted of 79 adult females (>3.5 years of age), 51 adult males (>4.5 years of age), 24 juveniles and 10 infants. Experiment 1 was conducted from December 2018 to April 2019, and Experiment 2 was conducted from January 2020 to May 2020. 

The macaques receive food twice daily with the first feeding occurring between 9:00am and 11:00am. Provisioned food consists of various fruits and vegetables, as well as wheat. The natural vegetation of the area provides the monkeys with additional foraging opportunities. Water is available ad libitum at any time from a natural stream that goes through the enclosure. Testing occurred between regular feeding sessions and did not involve subjecting the macaques to deprivation states. 

### 2.2. Ethical Note

The experiments took place in a wooden hut within the enclosure. The wooden hut and experimental apparatus had been used in a previous experiment [42], so the animals had already had an opportunity to habituate to its presence. Open doorframes were located at each corner of the hut to allow individuals to enter and leave the area at will. The macaques were neither actively selected, nor separated from the group. Data were collected on an opportunistic basis, with active participants and group size in the experimental area varying considerably across sessions. All participation of the macaques in our experiments was thus uncoerced. 

Since all our experiments were non-invasive, the study complied with the Austrian Law (§ 2. Federal Law Gazette number 501/1989) and the Code for Best Practices in Field Primatology and received oversight from and was authorized by the internal board of the Austrian Research Center for Primatology. No invasive research or experimental procedures requiring ethics approval according to the European Directive 2010/63 were performed. Our studies adhered to the American Society of Primatologists’ principles for the ethical treatment of primates, and all applicable international, national and institutional guidelines for the care and use of animals were followed.

### 2.3. Apparatus 

The apparatus used in our study (both Experiment 1 and 2) was a modified version of the string-pulling paradigm pioneered by Crawford [9] and later refined into the loose-string paradigm by Hirata [10,11]. The body of the apparatus consisted of a wooden platform with a clear lockbox mounted on top, in which food could be placed (Figure 1). A copper pipe that housed a wire ending in a loop extended from two opposite sides of the lockbox to an area in front of the apparatus. Each loop could be pulled to release one of the two pins within the lockbox. The floor of the lockbox contained a trapdoor mechanism that would drop open when both pins holding it in place were released. The experimenter was able to place food in the lockbox through a door in the top, but the only way for the monkeys to access the food was to spring the trapdoor by pulling both loops simultaneously. Successful release of the trapdoor resulted in the enclosed food being dropped onto the wooden platform below. The platform was positioned approximately 1 m away from the loops and could be approached from all directions, allowing for better observation of the impact of social dynamics on reward division. During both experimental phases, the loops were positioned two meters apart so that no single individual could reach both loops simultaneously (Figure 1).

### 2.4. Individual Training

Training consisted of two phases and was carried out from October to November 2018. Throughout the training period, individuals could succeed in releasing the food without a partner (see below). This allowed them to form an association between the apparatus and food acquisition while giving them the opportunity to learn how to operate it individually. For an individual to be considered trained, they needed to successfully complete at least three trials split between two separate days. Sessions were conducted up to three times daily and consisted of 10 trials. At the beginning of each trial, the experimenter locked two pieces of food within the apparatus before moving away to allow the monkeys clear access to the apparatus. Successful trial completion entailed pulling the active loop(s) to trigger the release of food. Once the food rewards were retrieved, the setup procedure was repeated until completion of the 10th trial marked the end of the session. During the first phase of training, only one loop was required to trigger the trapdoor and release the food (“single-sided pull” form), i.e., only one of the two pins holding the trapdoor was in place. The side actively required to operate the device was switched between the right and left side over the course of this phase to discourage perseveration of operation on only one side. This phase of training continued for 42 sessions.

During the second phase, which ran for 36 sessions, simultaneous pulling of both loops was required but the loops were positioned close enough together so that a single monkey could reach both loops at once (20 cm apart). The “simultaneous pull” form of the device was used only with monkeys who had met training criteria in the previous phase. If potential new pullers began manipulating the loops, the apparatus was switched back to the “single-sided pull” form of the previous phase. The “single-sided pull” form of the device was occasionally used with experienced monkeys as well as on an as-needed basis to maintain a high level of motivation.

By the end of the training period, 11 monkeys had reached training criteria on the “single-sided pull” form of the device, but no monkey had met the training criteria on the “simultaneous pull” form. Due to time-constraints, we decided to nonetheless proceed with Experiment 1. Note that Experiment 1 specifically tested for the monkey’s understanding of the cooperative task and allowed further learning of the contingencies of the set-up (see below). We ran an additional training phase consisting of 51 sessions in February 2019 to address a motivational issue that developed during the initial testing sessions. Two additional monkeys reached “single-sided pull” training criteria during this second training phase, bringing the total number of trained monkeys to 13. A further two learnt to use the apparatus during the testing phase of Experiment 1 without having participated in the training sessions.

The first 10 sessions of Experiment 1 were treated as a transitionary period for the dyadic form of the apparatus, with the experimenter acting as cooperative partner for any persistently interested individuals when no other potential partner was present. This was done to give individuals an opportunity to learn the necessity of a cooperative partner in the early stages of the setup and decrease the likelihood that they would lose interest in the apparatus before acquiring an effective cooperative partner. All of these sessions were, however, excluded from later analyses.

## 3. Experiment 1

The purpose of Experiment 1 was to investigate whether participants were capable of understanding the task outcomes associated with the cooperation paradigm. Special attention was given to demonstrating whether they understood the necessity of a partner. We ran a total of 126 testing sessions between December 2018 and April 2019. During these sessions, the loop handles were positioned two meters apart so that no single monkey could reach both loops simultaneously.

### 3.1. Procedure

Sessions consisted of 10 trials and were conducted up to three times daily, the first of which was conducted at least one hour after the morning feeding. In each trial, two pieces of food were locked into the apparatus to serve as rewards. For this purpose, we variously used apples, bananas, tomatoes and pineapples, all of which were known to be preferred foods for the monkeys [37]. The combination and presentation order of rewards was randomized. After loading the device, the experimenter moved to the edge of the research hut or left it entirely so as not to obstruct the monkeys’ access to the device or affect their behavior. As the monkeys were well habituated to the presence of humans, we anticipated the presence of the experimenter would have very little impact on their behavior during the experiment. If the food items were successfully retrieved from the apparatus, the experimenter returned to repeat the setup procedure until the session had ended. If a single dyad was monopolizing the device, defined as succeeding in six successive trials, the researcher would leave the experimental area for a short period of time (5–10 min) to encourage those individuals to leave and provide a new dyad with an opportunity to use the apparatus.

A trial was terminated if no monkey was successful in operating the device within 15 min unless there were monkeys actively interested in the device at that time in which case the trial continued until they succeeded or lost interest in the apparatus. If there was only one monkey actively interested in the device at the designated termination time, the experimenter attempted to act as a cooperative partner to give the individual an opportunity to gain experience with the device and to ensure that the behavior of persistent individuals was occasionally reinforced (see also section Training above). If a session continued for 1.5 h without reaching 10 trials, the session was terminated.

### 3.2. Measures: Collection and Coding

All sessions were fully recorded using cameras (Sony Handycam HDR-CX130E) mounted in protective boxes in two opposite corners of the research hut. The cameras captured two different frontal angles of the experiment, ensuring that the entire apparatus was within view. Trials were coded for identity of the pulling individuals, number of pulls (for definition see below), number and identity of the individuals in the surrounding area, whether a potential partner was positioned in front of the opposite loop at the time of the attempt, and whether the pulling individual waited for a partner to arrive before pulling. An attempt was defined as “any manipulation of the loop using enough force to result in movement of the pin to the open position”. Multiple pulls occurring within a 5-s span were aggregated and counted as a single attempt. A “potential partner” was defined as any individual who had previously succeeded in cooperation or met the first training criteria. An attempt was classified as a “wait” if it was preceded by the eventual puller delaying their pulling behavior for at least 15 s upon arriving to the area in front of the apparatus until such a time that a potential partner entered the area. All trials were coded live and afterwards from the video recordings. A second rater independently recoded 15% of the videos, and inter-rater agreement was excellent for participant identity (Cohen’s kappa = 0.98), potential partner presence (Cohen’s kappa = 0.97) and attempt (Cohen’s kappa = 0.91), and good for waiting (Cohen’s kappa = 0.71).

### 3.3. Analyses

To evaluate whether the Japanese macaques understood the specifics of the task, we ran three separate binomial generalized linear mixed models (GLMMs) with a logit link function on (a) whether individuals became increasingly proficient at cooperating, as reflected by the likelihood of an attempt to be successful, (b) whether individuals became more sensitive to the presence of a partner at the opposite loop, as measured by the likelihood of making an attempt with a potential partner present, and (c) whether waiting behavior increased in likelihood over time. Session number was entered as a fixed effect and number of attempts per individual was added as a random effect to control for repeated measurements. To correct for the alternative explanation that making an attempt with a partner present became more likely over the course of the experiment because the research hut increased in popularity, the number of individuals in the area was added to model b) as a fixed effect. We compared all full models to null models containing only the random effects. All statistical analyses were carried out using R statistical software (version 3.6.1) [43] with α set at 0.05. GLMMs were run using the lme4 package [44].

### 3.4. Results

Over the course of the experiment, we conducted 1165 trials spread across 126 sessions. Of these trials, 708 were successful, resulting in an overall success rate of 60.8% (708/1165). Twelve monkeys succeeded in cooperating in at least one instance (see Table 1). Four individuals who succeeded at least once in the training setup never engaged in cooperation and were excluded from further analyses. Of the cooperating individuals, only two were male. The age of participants ranged from 2 to 11, with an average of 6.1 (SD = 2.8). Participants had an average of 118 successes (SD = 183.7) and an average of 3.3 partners (SD = 2.8). Overall, these individuals combined into 20 unique cooperative dyads (Appendix A). Three of those dyads were composed of kin, and those three dyads accounted for 64.5% of all successful co-operations. Cooperative dyads had a mean of 35.4 successes (SD = 67.2).

Of the 126 sessions that were run, 10 sessions comprising the transitionary phase from the training to the experimental phase were excluded from all analyses as were an additional 12 sessions where problems with the videos made coding impossible. The co-operations that occurred during these sessions are still included in the descriptive summary tables.

Over the course of the experiment, individuals became more proficient at operating the apparatus with the likelihood of successful attempts increasing as a function of session (GLMM: Estimate = 0.398, z = 10.3, *p* = <0.001; Figure 2). This model explained the data significantly better than the respective null model (χ^2^(1) = 107.07, *p* = < 0.001).

The second model revealed a significant decrease in the likelihood of attempts being made in the absence of a potential partner over the course of the experiment (GLMM: Estimate = 0.155, z = 3.666, *p* = <0.001; Figure 3). This model explained the data significantly better than the respective null model (χ^2^(1) = 13.538, *p* = <0.001).

The final model revealed that the likelihood of waiting behavior did increase over the course of the experiment as a function of session (GLMM: Estimate = 0.378, z = 4.0, *p* = <0.001; Figure 4). This model explained the data significantly better than the respective null model (χ^2^(1) = 16.205, *p* = <0.001). Examination of total instances of waiting on the individual level revealed that waiting was used as a strategy by only a few individuals (Appendix A), suggesting that the main effect of waiting results from the behavior of only a minority of the participating individuals.

## 4. Experiment 2

The purpose of Experiment 2 was to assess whether the Japanese macaques involved in our study were sensitive to differing reward equity outcomes. To evaluate this, the food rewards used in the experiment were modified to create a condition where one participant always received a lesser reward than the other. A post-reward observation period was added to assess whether unequal divisions had any effect on the subsequent behavior of participants. For this experiment we ran a total of 119 sessions between January and May 2020.

### 4.1. Procedure

Other than the addition of the new reward division condition, the procedure used in Experiment 2 was identical to that of Experiment 1. The only further exceptions were that the experimenter under no condition acted as a cooperative partner for the monkeys and the trial termination time was shortened to 10 min.

As in Experiment 1, two food rewards were paired together in every trial. Both bananas and zucchinis were used as rewards, with reward type always matched within the trial. In Condition A, the two food rewards were sliced into segments 1/9th the size of the entire food item. In Condition B, a 1/9th-sized segment was paired with a 1/3rd-sized segment to ensure that the cooperating monkeys would receive unequal rewards. It should be noted that reward theft was always a possibility, so receiving no reward was a further possible outcome for the participating monkeys in either condition. Sessions alternated between Condition A and Condition B, such that each new session marked a change in condition.

### 4.2. Collection and Coding

At the beginning of each session, the experimenter moved into a corner and began voice recording (using an Olympus WS-510M voice recorder) any stress-related or aggressive behaviors occurring within the research hut using an ad libitum/behavioral sampling method [45,46]. Stress-related behaviors included scratch, yawn and body shake, while prototypical examples of aggressive behaviors were bite, chase and lunge (for a full list, see Appendix A). The reward division outcomes of each trial as well as any refusals to act as a cooperation partner were also noted. Reward division outcomes were classified in the following way: reward equivalent to that of the other recipient (Outcome A), reward smaller than that of the other recipient (Outcome B), reward larger than that of the other recipient (Outcome C) or no reward due to theft (Outcome D). A refusal was defined as “any instance where a focal animal that had previously cooperated in the session with an individual currently waiting near a loop does not act to cooperate with that individual for at least 10 s”. However, such a refusal was only observed 7 times over the course of 272 successful trials and was therefore excluded from future analyses.

To control for whether seeing an alternative reward available affected later behavior, whether an individual looked into the food box prior to cooperation was also recorded. Such instances were defined by an individual climbing up the apparatus to the plexiglass food box and looking inside. Sessions were video recorded using the same camera setup as Experiment 1, but in this experiment audio recordings from a handheld device served as the primary behavioral record so as to allow the experimenter greater mobility.

Following successful cooperation, post-cooperation behavior was recorded of one, or if possible, both members of the cooperative dyad. If one of the cooperating monkeys left the research hut following a successful trial, the session was paused, and a three-minute behavioral sampling period modelled after de Waal and Yoshihara’s post-conflict matched-control method [47] was conducted. In our modified form of their method, receiving an unequal reward in relation to a partner (Condition B) was treated as the conflict, whereas an equal reward distribution among the cooperation partners (Condition A) was treated as the control. A further difference was that monkeys were observed for three minutes instead of five because previous studies using the post-conflict method with Japanese macaques have found that the majority of noteworthy post-conflict behaviors in the species occur within the first two minutes following the conflict [48,49]. Furthermore, instead of using an observation of the same monkey on the following day as a matched control, our study compared the average behavior following an unequal outcome with the average behavior following an equal outcome.

If the cooperating dyad did not leave the research area after a successful cooperation, the next trial began immediately so as to maintain the interest of participants. If any of the remaining participants failed to show interest in the new trial, their behavior was categorized as post-cooperative behavior for three minutes following the successful cooperation.

### 4.3. Analyses

The initial data set was heavily inflated by zeroes, so measures expressing stress and aggression were additionally coded as binomial scores representing whether or not the behavior occurred (yes/no; with 1 as yes) and a hurdle approach was chosen for the analysis. The hurdle approach is a multi-step procedure in which an initial analysis deals with zero-inflated count data and a subsequent analysis examines only those cases with a positive count. Our initial hurdle began by comparing across reward division outcomes the likelihood of any stress-related or aggressive behaviors occurring within the post-cooperation observation period using a binomial GLMM with a logit link function. The second hurdle took only those observation periods where stress or aggression actually occurred and explored whether the intensity with which those behaviors occurred varied systematically in accordance with reward division outcome using a linear GLMM with a logit link function. The term “intensity” in this paper refers to the frequency of expression of the behavior of interest within a specified time period. Focal animal, partner, food type and observer were added to the models as random effects. Both models examined separately the first minute of data and the first three minutes of data following a successful cooperation to investigate whether any behavioral response was more pronounced immediately following the event. An estimated marginal means (EMM) function was used to compare the likelihood of expressing stress and aggression across outcomes.

A post-hoc analysis was done to see whether an individual that inspected the box before cooperating had a higher likelihood of expressing stress-related or aggressive behaviors after experiencing Outcome B. We used a binomial GLMM with a logit link function to test for the effect of looking in the box on the likelihood of expressing stress-related and aggressive behavior. The variables focal animal, food type and observer were added as random effects.

To determine whether the behaviors observed during the post-cooperation observation period were products of only the last trial outcome or an accumulation of all previous trials, another binomial GLMM with a logit link function was carried out to control for repeated measures. To check for a pattern each outcome was assigned a value: A, B, C and D, respectively, 1, 0, 2, −1. These values are arbitrarily assigned numbers used to make a distinction between the outcomes and are based on the cooperator’s direct personal food reward gains. Each trial had a value determined by the last outcome, and a cumulative value of the last and all previous outcomes of the individual during that session. Subsequently, the likelihood of stress-related and aggressive behaviors occurring within three minutes after the outcome was tested for the influence of the value of the last outcome and the “cumulative value of all previous trials” as a fixed effect. Consecutively, the same models were carried out with either only “cumulative value of all previous trials” or “direct value” as a fixed effect. In all models, focal animal was added as a random effect. ANOVA-based comparisons, using the Aikake information criterion (AIC) [50], between these models and null models were performed to see which model best explains the results. Since there were no cumulative values of previous trials for first trials per individual, only successful trials in which the individual had also succeeded in the previous trial were included in these analyses.

All statistical analyses were carried out using R statistical software [43] with α set at 0.05. GLMMs were run using the lme4 package [44] and EMM comparisons made use of the emmeans package [51].

### 4.4. Results

Over the course of the experiment, 119 sessions and 1059 trials were conducted, 272 of which were successful, resulting in an overall success rate of 25.7%. Seven monkeys succeeded in at least one cooperation, with four of those individuals accounting for 96% of all co-operations. Of the seven participants, six were returning monkeys from Experiment 1 while the seventh was a newly interested monkey who had shown no previous aptitude.

Breaking down the reward division outcomes observed across both conditions, in 41% of trials the cooperating monkey received an equal reward to that of their partner (Outcome A), in 11% of trials they received a smaller reward (Outcome B), in 12% of trials they received a larger reward (Outcome C), and in 41% of cases their reward was stolen (Outcome D).

#### 4.4.1. Stress-Related Behavior

We found a significant effect of the outcome of reward division on the likelihood of expressing stress-related behavior in the 3-min period following a cooperation (Table 2). Specifically, the cooperators were more likely to express stress-related behavior after Outcome B than Outcome A in the 3-min period following a cooperation (Estimate = −1.179, z.ratio = -2.888, *p* = 0.02), more likely to express stress-related behavior in Outcome D compared to Outcome A (Estimate = −1.296, z.ratio = −4.241, *p* = <0.001) and more likely to express stress-related behavior after Outcome D than Outcome C (Estimate = −1.254, z.ratio = −2.576, *p* = 0.049; Figure 5). This model explained the data significantly better than the respective null model (χ^2^(3) = 24.938, *p* = <0.001). We further found that looking in the box had no impact on the increased likelihood of expressing stress-related behavior following a personally disadvantageous inequitable division (Estimate = 0.263, z-ratio = 0.399, *p* = 0.690).

The same analysis was carried out using only the first minute of stress-related behavior following a cooperation, but no significant difference was found between the different reward outcomes.

The second hurdle, investigating the frequency of stress-related behavior when it occurred, found no significant difference in the intensity of stress-related behaviors (when the animals showed at least some stress-related behavior) following the different outcomes.

#### 4.4.2. Aggression

We found a significant effect of the outcome of reward division on the likelihood of expressing aggressive behavior in the 3-min period following a cooperation (Table 3). Specifically, a higher likelihood of expressing aggressive behavior after Outcome A compared to Outcome D in the 3-min period following a cooperation (Estimate = −1.189, z.ratio = −3.008, *p* = 0.014) as well as higher likelihood of expression in Outcome A compared to Outcome D (Estimate = −1.191, z.ratio = −4.09, *p* = <0.001; Figure 6). This model explained the data significantly better than the respective null model (χ^2^(3) = 20.735, *p* = <0.001). We further found that looking in the box had no impact on the increased likelihood of expressing aggressive behavior following an inequitable division (Estimate = −0.105, z-ratio = −0.173, *p* = 0.862).

The same pattern was present using only the first minute of data following successful cooperation, with a higher likelihood of aggressive behaviors again found in Outcome B compared to Outcome A (Estimate = −1.224, z.ratio = −3.094, *p* = 0.011) as well as higher likelihoods of aggressive behaviors in Outcome D compared to that of Outcome A (Estimate = −1.169, z.ratio = −3.983, *p* = <0.001), even after null model comparison (χ^2^(3) = 20.421, *p* = <0.001).

The second hurdle, investigating the frequency of aggression, when it occurred, found no significant difference in the intensity of aggressive behaviors following the different outcomes.

#### 4.4.3. Cumulative Effects

As for comparing the models of the effect of cumulative value of all previous trials on the likelihood of expressing stress-related and aggressive behavior, we ran binomial GLMMs with logit link functions including and excluding the effect of cumulative reward outcome. The GLMM testing the effect of the value of the last trial on likelihood of stress excluding cumulative outcome value as a fixed effect revealed a significant effect of the last trial (Estimate = −0.698, z-ratio = −4.200, *p* = <0.001). The GLMM including cumulative outcomes as a fixed effect did not reach significance (Estimate = 0.097, z-ratio = 1.284, *p* = 0.199) nor did it when it was tested as a single fixed effect, i.e., without the value of the last trial (Estimate = -0.0833, z-value = −1.312, *p* = 0.189). For the likelihood of aggressive behavior, the effect of value of the last trial without the cumulative value of all previous trials as a fixed effect was significant (Estimate = −0.463, z-ratio = -3.384, *p* = <0.001), while the GLMM including cumulative outcomes as a fixed effect did not reach significance (Estimate = 0.0120, z-ratio = 0.177, *p* = 0.860), nor did it when it was tested as a single fixed effect (Estimate = −0.094, z-value = −1.634, *p* = 0.102). ANOVA-based comparisons revealed that the model where stress likelihood was tested as a result of the value of the last trial without additional fixed effects had the lowest AIC (265.64) compared to when only the effect of all previous trials was tested (AIC = 284.48), when the effect of all previous trials and that of the last were tested (AIC = 265.99) and the null model (AIC = 284.23). The same applied to the likelihood of aggression, where the model with effect of only the last trial had an AIC of 328.48. For comparison, the model testing only the effect of all previous trials had an AIC of 337.97, while the model including both previous trials and the last trial as fixed effects had an AIC of 330.45 and the null model had an AIC of 338.69. This led us to the conclusion that the likelihood of post-cooperation stress-related and aggressive behavior was better explained by the outcome of the last trial than it was by the cumulative value of all previous trials.

## 5. Discussion

Our study used a variation of the loose-string paradigm [10,11] to examine cognitive abilities associated with cooperation in Japanese macaques. Even though it has been previously demonstrated that the species is capable of succeeding in a cooperative task [14], we took this a step further by demonstrating that they are attentive to the presence of a partner. In addition, we combined our cooperation task with an examination of the reaction to unequal payoffs thereafter. Embedding our experiments within a setting resembling the natural environment of the species allowed for an examination of cooperation and inequity aversion (IA) within an intact social environment. This allowed us to place IA in the context of natural reward division following cooperation, thus more closely replicating the conditions that should accompany its occurrence in nature [52]. Our study used post-cooperation observation of stress-related and aggressive behaviors to investigate the potential existence of IA in the species, and succeeded in documenting a pattern of behavior consistent with the predicted profile of disadvantageous IA. That being said, we did not provide evidence of the adoption of IA-associated behavioral strategies such as future refusal to cooperate or partner switching (cf. [53]). Future studies may wish to focus specifically on how the negative states suggested by the behavioral profile we observed in our study may translate into strategies that could conceivably result in consistent positive fitness outcomes.

### 5.1. Sensitivity to Partner Presence

The primary aim of our first experiment was to demonstrate that Japanese macaques recognize the importance of their partner during a cooperative task. Demonstrating such an understanding is an essential step in arguing a case for intentional cooperation on the part of the macaques. We assessed this by examining changes in the proportion of attempts made in the presence of a potential partner as well as changes in the frequency of waiting behaviors. Our analyses demonstrated a behavioral shift in the predicted direction for both features, with a decrease in the proportion of attempts made without a potential partner present and an increase in the frequency of waiting behavior. This led us to the conclusion that at least some of the individuals involved in our experiment developed an understanding of the contingencies of the task, including the necessity of having a partner, though they may not have understood the role that their partner played in the task. This corresponds to “presence-dependent cooperation” in the framework set out by Albiach-Serrano [8], though the level of understanding in our study population could have extended beyond this.

Our experiment further demonstrated that individuals became increasingly proficient with more experience, with successful attempts increasing in relative frequency to unsuccessful attempts as the experiment progressed. Even though it is not a strong argument for task understanding in and of itself, when taken in conjunction with the increasing specificity in which participating monkeys made attempts, it bolsters the case for task understanding in at least some individuals. Increasing relative frequency of successful attempts is suggestive of greater coordination with cooperative partners.

Even though increasing specificity of pulling was evident in the case of most regular cooperators, this does not necessarily indicate that all individuals who showed this pattern understood the cooperative nature of the task. For a dyad to give the appearance of deliberate cooperation, only one member of that dyad needs to be actively coordinating with the other. Increasing specificity of pulling as we measured it may just as easily indicate that another individual became adept at coordinating with them, rather than that they became proficient at coordinating with others. Used as a proxy for task understanding, increasing specificity of pulling needs to be interpreted with caution when taken in isolation. There was a significant main effect for an increase in the frequency of waiting behavior as well, but closer examination revealed that this behavioral strategy was only frequently used by three individuals (Appendix A). Given that all but two of the dyads that formed contained at least one of these individuals, it is a distinct possibility that all active coordination throughout the experiment rooted from them. It is worth noting that these three individuals were also the youngest monkeys involved in the experiment. Nevertheless, active coordination even in the case of only a few individuals still demonstrates a capacity for task understanding and deliberate cooperation in the species, thus serving as proof of concept.

Imposing a delay in partner release has been effectively used to demonstrate waiting behavior during cooperation tasks in a variety of species including Asian elephants (*Elephas maximus*) [19], capuchin monkeys (*Cebus apella*) [54], cotton-top tamarins (*Saguinus oedipus*) [55], chimpanzees (*Pan troglodytes*) [12], spotted hyenas (*Crocuta crocuta*) [21], domestic dogs (*Canis familiaris*) [56], orangutans (*Pongo pygmaeus*) [57], coral trout (*Plectropomus leopardus*) [58] and keas (*Nestor notabilis*) [24], but the practical constraints of the semi-free setting used in our study prevented us from designing our own experiment in this way. Even though naturally occurring waiting was observed in a few of our monkeys, our argument for sensitivity to partner presence rests more heavily on the change in the proportion of attempts made in the presence of a partner.

When it comes to behavioral inhibition, macaque species have been demonstrated to underperform in comparison to other primates [59,60], which may have impaired their capacity to wait for a partner. Indeed, only one of the Japanese macaques in Kaigaishi, Nakamichi, and Yamada’s previous study of the species showed any indication of using waiting as a strategy [14]. Evans and Beran, in their study of the closely related rhesus macaque (*Macaca mulatta*), found that the species had a particularly difficult time maintaining self-control over a delay when an impulsive response option was continually available [60], as it was in the present study. An absence of waiting due to a lack of inhibitory ability in a species does not necessarily mean that the individuals involved do not have some understanding that a partner is necessary to complete the task, it may merely reflect the behavioral constraints intrinsic to the species. It was with this in mind that in addition to examining waiting behavior, we chose to emphasize the more subtle measure of change in proportion of attempts as a measure of sensitivity to partner presence in our own study.

An alternative explanation for the increase in the relative frequency of pulling behaviors in the presence of others is the possibility of social facilitation, a well-documented phenomenon in which a dominant behavioral response is amplified in the presence of others [61]. Our argument against the social facilitation explanation is as follows: firstly, the effect of social facilitation should have been more or less constant across all individuals who came to associate pulling behavior with reward, but this was not the case. Some individuals that were highly active in the experiment did not increase at all in relative frequency of pulling behaviors in the presence of others. One of the most successful individuals even showed a decrease by this metric over the course of the experiment, which is more in line with our assertion that only a few individuals gained insight into the cooperative nature of the apparatus and adjusted their behavior accordingly. Furthermore, as training sessions took place before the experimental phase began, pulling behavior should have already been the dominant response at the beginning of the experiment and remained the dominant response throughout. Any effect of social facilitation should therefore have remained constant.

As for why pulling in the absence of a partner never fully extinguished, the fact that our experimental apparatus automatically reset itself after failed attempts likely played a large part. In the typical loose-string paradigm, an attempt made in the absence of a partner results in an inability to further participate in the experiment until the apparatus is reset by an attendant. With the apparatus used in our experiment, repeated attempts could be made without precluding the possibility of future attempts should a partner appear. In practical terms, this meant that the only cost of making an attempt on the device was the small energy input required of the attempt. This cost was likely low enough that the pressures acting toward the extinction of indiscriminate pulling were not strong enough to extinguish that behavior outright. The benefit of designing our experiment in this way is that it allowed for individuals not met with immediate success to continue exploratory behavior with no ill effect until a potential partner appeared and gave them an opportunity to learn the appropriate reward contingency. This was well-suited to the natural setting of our study, as long periods of time may pass before a potential partner arrives.

### 5.2. Partial Evidence for Inequity Aversion

Our second experiment explored whether Japanese macaques react aversively following cooperation that has resulted in an unequal reward outcome. Our results demonstrate that the subjects in our study were more likely to display stress-related and aggressive behaviors after receiving a disadvantageous reward outcome in comparison to another individual (Outcome B) than when receiving an equally sized reward, but in this condition also equal to that of their partner (Outcome A). The higher likelihood of expressing stress-related and aggressive behavior in the post-cooperation period following Outcome B does indicate that a personally disadvantageous outcome following cooperation incites aversive behaviors, which may be caused by IA. The lack of a difference in the likelihood of expressing stress-related behavior and aggression between Outcome C, i.e., when the subject received the larger of the two rewards, and Outcome A, suggests that this cannot be explained by the frustration of individuals over the fact that they received a lesser reward after having received the larger reward previously [62,63,64]. We cannot completely exclude, however, the possibility that the reaction of the Japanese macaques was due to the violation of expectation of receiving the visible larger rewards [65,66,67,68,69,70]. Even though the available rewards were visible before reward division occurred, any interested macaque had to move directly up to the plexiglass food box to see them. We included whether this occurred in our analysis and found that it had no impact on the increased likelihood of expressing stress-related or aggression following an inequitable division. Consequently, we concluded that a social comparison of rewards was most likely responsible for the pattern that we observed.

Engelman et al. presented another explanation for disadvantageous IA in the form of the “social disappointment hypothesis” [71], in which an individual rejects a reward because it is disappointed in the provider for not supplying a better reward. Their findings suggest that reward rejection may be based on a social interaction with the researcher, rather than a comparison with the reward of the conspecific, because they only found rejections of those rewards when provided by a researcher and not when provided by a machine. Even though a researcher was present during the experiments in the current study, our method distinguishes itself from the researcher condition of the study of Engelman et al. [71] in that in our experiment, the researcher did not distribute the food rewards. Instead, the rewards became simultaneously available to all individuals after joint completion of the dyadic task. It must be noted, however, that the macaques could see the researcher loading the device with food. Nevertheless, as the specifics of the reward division were left to the macaques, they could at most be disappointed in the experimenter for loading the apparatus with unequal food rewards. Such inter-conspecific negotiation tasks with minimal researcher interference are suggested by Oberliessen and Kalenscher to control for the “social disappointment hypothesis” [72].

Finally, a comparison of the outcomes is not the only important factor in fairness sensitivity—comparison of the efforts is also of relevance [35,73,74,75,76]. One could argue that, in the context of the current study, a thief has a lower effort input than a cooperator. It is then of some relevance that the macaques in our study showed no difference in their likelihood to react aversively whether a thief or their cooperative partner obtained the second reward (Appendix A). This suggests that macaques may not base their aversive reaction on a comparison of effort when obtaining a reward. However, the comparison between the role of thief and cooperation partner was not one of the premeditated goals of this study, and there are methods better suited for comparing effort-reward (in)equity [35,74,75,76]. Future studies that aim to research effort (in)equity should take advantage of these more direct methods.

In general, the proximate mechanism underlying IA are still poorly understood, but frustration and anger, as proxied here by stress behavior and aggression, seem potential candidates for an affective component involved in IA. Consequently, our findings provide tentative support for the theory that an aversion for inequity, potentially mediated by affective responses [6], is necessary to stabilize cooperation [77,78,79,80] and that cooperation and a sense for fairness have coevolved [31,77,78,79].

Our experiment distinguishes itself from the bulk of animal inequity studies in that subjects perform a cooperation task instead of conducting a version of the impunity game, which often translates into a token-exchange experiment where the subject must exchange a token with a researcher or machine in order to receive a reward (Capuchin monkeys (*Cebus apella)* [53,74,81,82], crows (*Corvus corone*) and ravens (*Corvus corax*) [75], chimpanzees (*Pan troglodytes*) [71,83], keas (*Nestor notabilis*) [24], Goffin’s cockatoos (*Cacatua goffiniana*) [76]). As it has been argued that IA has coevolved with cooperation [31,77,78,79], our cooperation experiment thus more closely resembles the context that may have created the selection pressures leading to the evolution of IA. Nevertheless, the use of more artificial set-ups may provide clearer indications of IA. In the impunity game, a subject can choose whether to continue performing a rewarded task after seeing a conspecific receive a greater reward for the same task (though their own refusal has no impact on whether that conspecific receives future rewards). Refusing to continue the task is seen as a potential indicator of IA in the species. In our current study, refusal to continue cooperating meant that neither the refuser nor their partner were able to obtain a reward, making our experimental setup more closely resemble that of the ultimatum game [84]. Despite the potential impact it could have had on an inequitable partner, refusal to cooperate was seldom observed in our experiment. It should be noted however, that as a reaction an individual could also simply leave the cooperation device. Yet, it was impossible to distinguish the different potential reasons for an individual to leave the cooperation device in our open setting. Nevertheless, this reaction should be investigated in more detail in future studies that also try to embed an examination of IA in a more natural cooperative situation.

The current study found that the likelihood of expressing stress-related behavior or aggression did not increase with the number of previous trials in which a monkey had personally experienced inequity. As our results suggest that macaques base their reaction only on the last reward-division outcome, our second hypothesis must be rejected. Brosnan suggests that individuals should react aversively to personal inequity primarily when it becomes a systematic outcome [31]. This sort of reaction profile allows for some tolerance of the accidental inequities that happen frequently in natural settings, which serves to maintain cooperative bonds that have overall equal payoffs. However, reacting to cumulative inequity rather than direct inequity may be reserved for species that cooperate in the food domain, where differences in obtained food after a cooperative task are more readily quantifiable. Even though IA in its basic form may not be domain specific [35], the cognitive accounting mechanisms required of cumulative IA may be. This ability would not necessarily be of use in the social domain, where outcomes are not as readily measurable. This may lead species that cooperate in only the social domain to employ a more cognitively simplistic mechanism such as only considering the immediately previous outcome. Japanese macaques may fall into this category, as they cooperate in the social domain and have an extensive network of kin [85] but are not known to hunt together or share food.

### 5.3. Limitations and Future Directions

An added obstacle when conducting research in a natural setting is the difficulty of obtaining a high rate of participation in the population [86]. Even though there are many advantages to conducting experiments in an intact social environment, these experimental designs come with the disadvantage of potential active interference from other group members and monopolization of the experiment by high-ranking individuals. This may be especially relevant in the study of socially despotic species such as the Japanese macaques used in our study. Furthermore, the alternative foraging opportunities available in a natural setting may reduce the appeal of food-rewarded experiments. Even though our study population contained 164 monkeys at the beginning of our first experiment, only 12 of them participated in it. Participation in our second experiment dropped down to seven monkeys, with only four of them accounting for 96% of all co-operations. Due to this fact, our conclusions need to be interpreted with caution. Future studies conducting research in a natural setting may wish to consult Cronin et al.’s recommendations for countering the reduced participation typical of such studies [86].

The monkeys in our study tended to have very few partners, which may be a further effect of the low level of social tolerance typical of the species under investigation [87]. Most individuals cooperated with a very small proportion of the other participants, and some of the monkeys who reached training criteria during the training phase never came to cooperate with any others despite a continued interest in the experiment. The juveniles in our experiment tended to have a much higher number of partners, constituting a notable exception to this pattern. This may be due to the greater social tolerance characteristic of juveniles in primate species [88]. Even though our sample size was too small to draw any meaningful conclusions regarding partner preference, the dyadic success rates we observed were suggestive of a strong preference for kin-kin or juvenile-juvenile dyads, as is expected of a socially despotic and nepotistic species. Future studies making use of a larger sample size may be able to shed further light on partner preference in socially despotic species such as the Japanese macaque.

Our limited sample in Experiment 2 further prevented us from investigating the impact of rank on reaction to reward division. Half of the monkeys involved in that experiment were sub-adult males, which occupy a particularly transient position in the hierarchy system of Japanese macaques [89,90,91]. All active dyads in this experiment included at least one of these males, so an analysis of the impact of rank on reaction to inequity could not be performed. This is nevertheless an interesting avenue of study, as past experiments have demonstrated that dominance plays a large role in access to food sources [92,93,94,95], and future studies may wish to revisit this subject in the species.

Even though we did make a case for “presence-dependent cooperation” in our study, we did not go so far as to investigate “action-dependent cooperation”, the next step in demonstrating fully intentional cooperation [8]. Past studies have used partner-directed glancing behavior as an indicator of attentiveness to partner action [54,57,96]. Our own study design could potentially be adapted to take glancing behavior into account using the same approach we used in making a case for sensitivity to partner presence. In addition to tracking the actions of a cooperation partner, active signaling may also contribute to deliberate coordination. Japanese macaques are extremely vocal animals [97], and it would be of further interest to investigate whether they use these vocalizations to recruit cooperation partners or otherwise coordinate their cooperative efforts.

## 6. Conclusions

We demonstrated not only that Japanese macaques are capable of succeeding in a cooperation task, but that they are also attentive to the presence of a partner and will actively adjust their attempt rate to match this. Over the course of the first experiment, the proportion of attempts made in the absence of a partner decreased and a subset of the participants learnt to wait for a partner when none was present. Our second experiment found that individuals had a higher likelihood to express stress-related and aggressive behaviors after receiving a lesser reward than their partner in the cooperation task when compared to a control condition where both individuals received an equivalent reward. This matches the pattern of behavior predicted by disadvantageous IA, suggesting a possible sensitivity to inequity in the species. A major strength of our study was that it combined a cooperation paradigm with a measure of IA within an intact social environment to better replicate food division dynamics that would be found in nature. This design leant a high degree of ecological validity to our experiments, which may be of particular importance for studies examining socially bound phenomena such as cooperation and IA.

## Figures and Tables

**Figure 1 animals-11-01497-f001:**
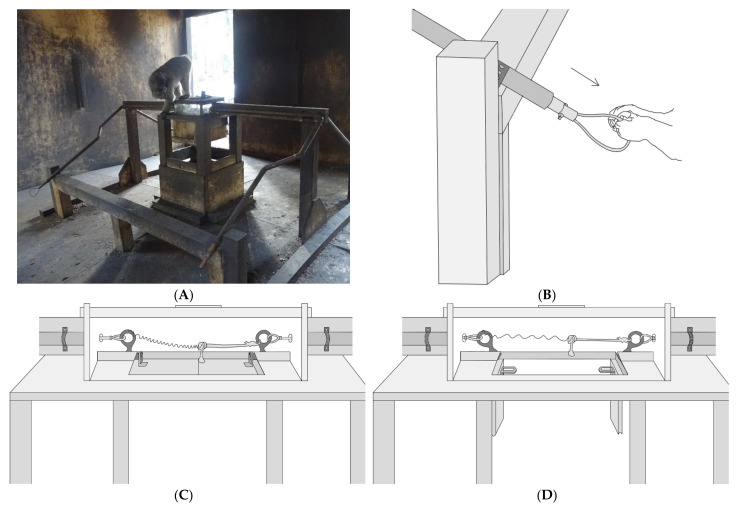
(**A**) Photo of the apparatus used in the study. Food rewards remain visible but inaccessible in the plexiglass lockbox until simultaneous pulling of the loops positioned at the end of the copper pipes triggers the opening of the trapdoor, allowing the rewards to fall onto the platform below. (**B**) Close-up illustration of one of the loops. (**C**) Close-up illustration of the lockbox in the closed position. A latch mechanism holds the trapdoor in place until both loops are pulled simultaneously, releasing the latch mechanism and allowing the trapdoor to fall open. (**D**) Close-up illustration of the lockbox in the open position. The trapdoor has fallen open after release of the latch mechanism.

**Figure 2 animals-11-01497-f002:**
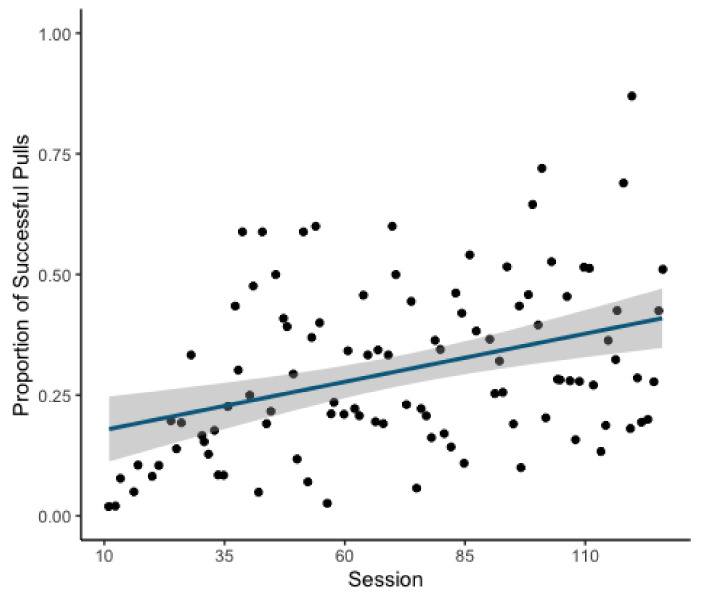
Scatterplot of proportion of successful pulls by session with line of best fit overlaid. Greyed area indicates standard error.

**Figure 3 animals-11-01497-f003:**
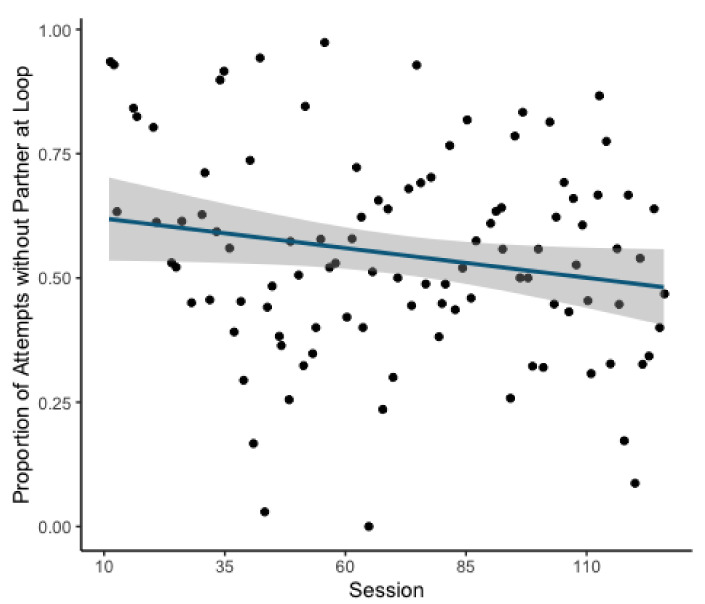
Scatterplot of overall proportion of attempts without a potential partner in front of the opposite loop. Line of best fit overlaid with surrounding greyed area indicating standard error.

**Figure 4 animals-11-01497-f004:**
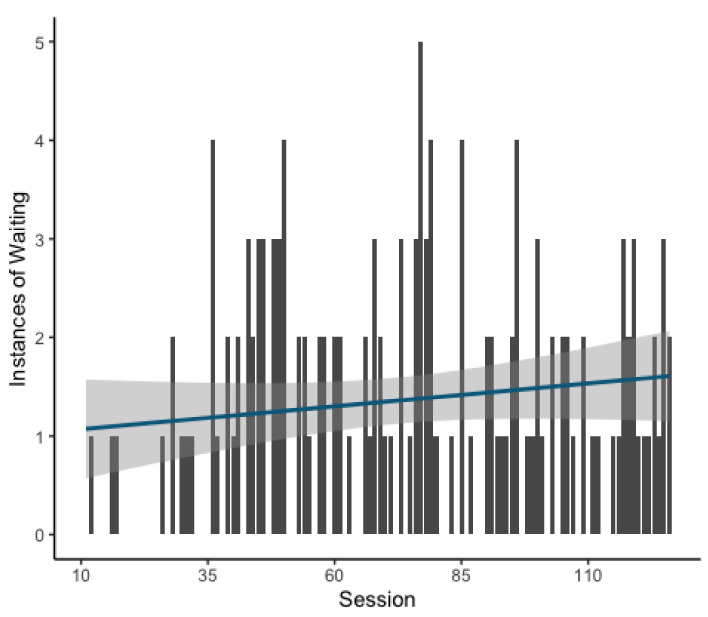
Bar graph showing instances of waiting per session with line of best fit overlaid. Greyed area indicates standard error.

**Figure 5 animals-11-01497-f005:**
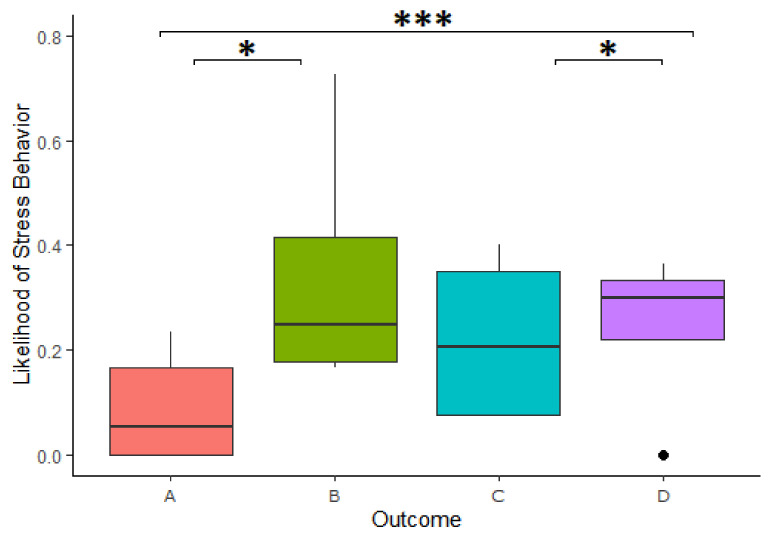
Likelihood of stress-related behaviors across outcomes. A = reward equivalent to that of other, B = reward smaller than that of other, C = reward larger than that of other, D = no reward due to theft. Boxplots show the median (solid line), 25th–75th percentile (box) and the largest and smallest value (whiskers). Dots reflects outliers. * indicates a difference at the *p* < 0.05 level and *** indicates a difference at the *p* < 0.001 level.

**Figure 6 animals-11-01497-f006:**
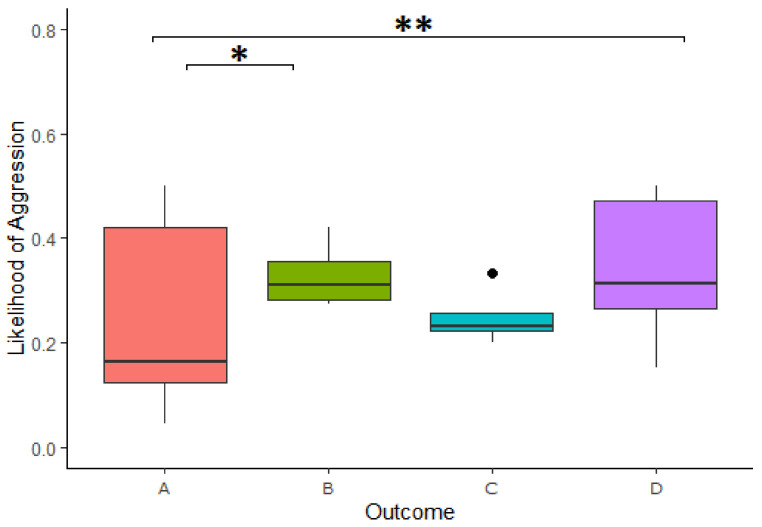
Likelihood of aggressive behaviors across outcomes. A = reward equivalent to that of other, B = reward smaller than that of other, C = reward larger than that of other, D = no reward due to theft. Boxplots show the median (solid line), 25th-75th percentile (box) and the largest and smallest value (whiskers). Dots reflects outliers. * indicates a difference at the *p* < 0.05 level and ** indicates a difference at the *p* < 0.01 level.

**Table 1 animals-11-01497-t001:** Summary of results by individual.

Individual	Age *	Sex	Partners	Total Successes
James	2	Male	11	635
Ingrid	10	Female	3	230
Pippi	2	Female	6	203
Kate	7	Female	3	177
Finn	5	Male	4	78
Alina	11	Female	2	58
Herta	6	Female	3	14
Montana	8	Female	1	10
Krato	5	Female	2	5
Sandra	7	Female	3	3
Julia	6	Female	1	2
Kurtney	4	Female	1	1

*: Age in years as of the beginning of the experiment.

**Table 2 animals-11-01497-t002:** Results of the GLMM testing the effect of outcome on likelihood of expressing stress-related behaviors within 3 min after cooperation.

Outcome	Estimate	Standard Error	z-Value	*p*-Value
A	−2.111	0.305	−6.918	<0.001
B	1.179	0.408	2.888	0.004
C	0.042	0.516	0.081	0.935
D	1.296	0.306	4.241	<0.001

**Table 3 animals-11-01497-t003:** Results of the GLMM testing the effect of outcome on the likelihood of expressing aggressive behavior within 3 min after cooperation.

Outcome	Estimate	Standard Error	z-Value	*p*-Value
A	−1.865	0.311	−6.008	>0.001
B	1.189	0.395	3.008	0.003
C	0.467	0.427	1.093	0.275
D	1.191	0.291	4.090	>0.001

## Data Availability

The data presented in this study are available in the Appendix A.

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
