# Peer review of "Exploring the Cognitive Capacities of Japanese Macaques in a Cooperation Game"

_animals, 2021, doi:10.3390/ani11061497_

Round 1

Reviewer 1 Report

General Comments

The manuscript provides original data pertaining to cognitive abilities in captive Japanese macaques through novel methodological approaches to the study of cooperation. As the authors correctly indicate, this work has the potential to add useful information to our understanding of macaque cognitive abilities underpinning their cooperative behaviour. This study may generate an interesting paper, and the manuscript has already very good structure and scientific soundness. I believe, however, it would benefit from some amendments to improve clarity and conciseness, as well as to avoid overselling the study findings.

Major Remarks

I believe there are some weaknesses to address in order to improve the manuscript:

  • The Introduction can be improved in terms of streamlining and clarifying the study hypotheses and predictions.
  • The description of Methods & Results of both experiments (3. Experiment 1 / 4. Experiment 2) should be streamlined, which would help to enhance the clarity and conciseness of the overall manuscript.
  • I would strongly advise to re-write the Discussion & Conclusions. The first paragraph of the Discussion is currently overselling the limited results of this study, while the remaining sub-sections should be rationalized. The Conclusions are also overselling the study findings, whilst future perspectives should be expanded.

Minor Remarks

Please see below my comments:

  • Abstract
    • Line 29: Delete “not in a vacuum, but”
    • Lines 30-32: Reword the sentence to make it more concise and without trashing the work of other authors
    • Line 33: Add sample size (n=?)
    • Line 33: Replace “a population of Japanese macaques (Macaca fuscata) living under semi-naturalistic conditions” with “a captive population of Japanese macaques (Macaca fuscata)”
    • Lines 34-37: I would advise to streamline, or even delete, these sentences to enhance conciseness of the abstract
    • Line 39: Replace “came to show” with “showed”
    • Line 42: Replace “possess not only some sort of” with “have some”
    • Line 43: Replace “but also” with “and”
    • Line 44: Reword “that occurs thereafter suggestive of inequity aversion”
    • Line 44: Delete “,and importantly, that this can be tested in an 44 ecologically valid context”.
  • Introduction
    • Lines 101: Delete “,and for the duration of this paper,”
    • Lines 105-106: “complex social structures” - further details needed
    • Line 109: replace “Massen et al.” with “it was”
    • Line 116: Delete “Here,”
    • Line 116: “living under semi-naturalistic conditions” – add definition & citation
    • Lines 119-127: Delete these sentences
    • Lines 135-139: Delete these sentences
    • Lines 116-153: Potential for improvement in terms of clear and concise study aims, hypotheses & predictions
  • Methods / Results
    • Line 155: Replace with “Study Subjects & Housing”
    • Line 158: “living under semi-naturalistic conditions” – please add a (convincing) definition and related citation… Do authors know the natural conditions of wild Japanese macaques? It looks to me as captivity within a large enclosure
    • Line 163: Delete “,” after “2018”
    • Line 166: Replace with 9:00am to 11:00am
    • Lines 176-179: Reword these sentences – these are currently more suitable to a study with human participants
    • Lines 180-186: Add reference to compliance to the relevant EC Directive
    • Line 207: Delete “2018”
    • Line 229: Check the typo(s)
    • Line 237: Delete the commas before/after “2019”
    • Lines 242-253: This paragraph is not necessarily informative/relevant, at least in its current form – delete or streamline it
    • Lines 287-289: Add details about camera (brand & model) and coding (software & version)
    • Line 289: Delete “individual or”
    • Line 305: Replace “came to understand” with “understood”
    • Line 315: “research area” – clarify
    • Line 373 (and throughout the manuscript): Authors have used some time “experimenter” and some time the author initials – need for consistency (i.e. always or never initials of individual experimenters)
    • Line 387 (and above): Ad libitum needs to be written in italics (as it comes from Latin)
    • Line 387: Add further citation to Altman (1974)
  • Discussion / Conclusions
    • Line 570: Which studies? Add citation(s)
    • Lines 570-577: Reword these sentences – authors should not oversell their study findings
    • Line 609: Replace “came to understand” with “understood”
    • Line 612-613: Reword – “became good” is not formal writing style…

Author Response

Reply to Reviewer 1

The manuscript provides original data pertaining to cognitive abilities in captive Japanese macaques through novel methodological approaches to the study of cooperation. As the authors correctly indicate, this work has the potential to add useful information to our understanding of macaque cognitive abilities underpinning their cooperative behaviour. This study may generate an interesting paper, and the manuscript has already very good structure and scientific soundness. I believe, however, it would benefit from some amendments to improve clarity and conciseness, as well as to avoid overselling the study findings

We would like to thank the reviewer for their many insightful suggestions. In addition to addressing the issues pointed out in the thorough line-by-line comments, we have made considerable amendments throughout the paper to increase its overall conciseness and clarity. We have also considerably downplayed our conclusions so as not to oversell our study findings.

I believe there are some weaknesses to address in order to improve the manuscript:

The Introduction can be improved in terms of streamlining and clarifying the study hypotheses and predictions.

A considerable amount of material from the last few paragraphs of the intro was deleted or rephrased, increasing the overall clarity of our message (ll.196ff).

The description of Methods & Results of both experiments (3. Experiment 1 / 4. Experiment 2) should be streamlined, which would help to enhance the clarity and conciseness of the overall manuscript.

In addition to attending to all of your smaller suggestions for rewording and deletion, we have streamlined our material in a few other points (l.422, l.433). Nevertheless, we have also chosen to keep several parts in the methods section which the reviewer thought were redundant, because we aim to be as transparent about the choices in our procedures as possible, as to allow others to be able to replicate our study.

I would strongly advise to re-write the Discussion & Conclusions. The first paragraph of the Discussion is currently overselling the limited results of this study, while the remaining sub-sections should be rationalized. The Conclusions are also overselling the study findings, whilst future perspectives should be expanded.

The first paragraph and conclusion of the discussion (ll.784ff, l.1166, ll.1170ff) have been altered considerably. Both were downsized in length and the claims made in them have been downplayed to avoid overselling our results.

We additionally renamed our limitations section “Limitations and future directions” (l.1108), making specific reference to how our limitations could be improved in future studies (ll.1120ff, ll.1133ff, l.1149) as well as adding an additional two paragraphs of content (ll.1136ff).

We do not fully understand what the reviewer refers to with rationalizing our subsections, as we do already provide clear rationale for each subsection; i.e. discussing alternative hypotheses and situating our findings in the existing literature.

Minor Remarks

Please see below my comments:

Abstract

The abstract as a whole has been reduced to about 3/4s its original size, which considerably increased its conciseness.

Line 29: Delete “not in a vacuum, but”

Done.

Lines 30-32: Reword the sentence to make it more concise and without trashing the work of other authors

We have completely rewritten the indicated sentences (ll.29ff). We feel this rewording more directly presents the theme of our study without straw manning other authors, thank you for pointing this out, it was never our intent to deride the work of others.

Line 33: Add sample size (n=?)

Added.

Line 33: Replace “a population of Japanese macaques (Macaca fuscata) living under semi-naturalistic conditions” with “a captive population of Japanese macaques (Macaca fuscata)”

We have instead reworded this to “a semi-free population of provisioned Japanese macaques” (l.32). We are hesitant to use the word “captive” (though it is technically correct) because it may summon up in our readers the wrong image of the living conditions of the monkeys. We now specifically clarify in the methods how the enclosure resembles their natural environment and have switched to using the term “semi-free” throughout, as many studies conducted under similar conditions have also used this term (Setchell et al., 2013; Naud et al., 2016; Ottoni & Izar, 2008; Blomquist et al., 2011; Nunn & Deaner, 2004; Trapanese et al., 2020; Colemann et al., 2011).

Lines 34-37: I would advise to streamline, or even delete, these sentences to enhance conciseness of the abstract

The indicated sentences have been reduced to a single sentence roughly half the length (ll.33ff).

Line 39: Replace “came to show” with “showed”

Done.

Line 42: Replace “possess not only some sort of” with “have some”

Done.

Line 43: Replace “but also” with “and”

Replaced with “as well as” (flowed better).

Line 44: Reword “that occurs thereafter suggestive of inequity aversion”

Changed to “suggestive of an aversion to inequity”.

Line 44: Delete “,and importantly, that this can be tested in an 44 ecologically valid context”.

Done.

Introduction

Lines 101: Delete “,and for the duration of this paper,”

Done.

Lines 105-106: “complex social structures” - further details needed

We have now specified the relevant social feature (coalition formation with non-kin for social support during conflicts) with citations (ll.186ff).

Line 109: replace “Massen et al.” with “it was”

Sentence has been rephrased, resulting in the removal of that phrase (l.188).

Line 116: Delete “Here,”

Done.

Line 116: “living under semi-naturalistic conditions” – add definition & citation

We have rephrased the sentence, providing a brief explanation behind our terminology with citation (ll.196ff) and further clarification provided in methods (ll.225ff).

Lines 119-127: Delete these sentences

Reduced to a single sentence less than half the length of the deleted material (ll.200ff).

Lines 135-139: Delete these sentences

Deleted.

Lines 116-153: Potential for improvement in terms of clear and concise study aims, hypotheses & predictions

After following your recommendations and deleting a few other unnecessary sentences (l.213, l.217), the word count of these paragraphs was reduced from 470 to 309. The resulting section is much more clear and concise, thank you for your suggestions.

Methods / Results

Line 155: Replace with “Study Subjects & Housing”

Done.

Line 158: “living under semi-naturalistic conditions” – please add a (convincing) definition and related citation… Do authors know the natural conditions of wild Japanese macaques? It looks to me as captivity within a large enclosure

As previously mentioned, we have changed the wording of this phrase to “semi-free”, and added further clarification regarding our usage of the term, with citation for comparison to natural environment (ll.224ff)

We thank you for the recommendation and hope that this alteration is to your liking.

Line 163: Delete “,” after “2018”

Done.

Line 166: Replace with 9:00am to 11:00am

Done.

Lines 176-179: Reword these sentences – these are currently more suitable to a study with human participants

The sentences have been suitably reworded (ll.344ff).

Lines 180-186: Add reference to compliance to the relevant EC Directive

Done.

Line 207: Delete “2018”

Done.

Line 229: Check the typo(s)

Fixed.

Line 237: Delete the commas before/after “2019”

Done.

Lines 242-253: This paragraph is not necessarily informative/relevant, at least in its current form – delete or streamline it

We’d like to retain parts of this paragraph for the sake of full transparency, but we have streamlined it to half the length (ll.417ff).

Lines 287-289: Add details about camera (brand & model) and coding (software & version)

Camera details have been added, but no coding software was used in our study so that wording is left as is.

Line 289: Delete “individual or”

Done.

Line 305: Replace “came to understand” with “understood”

Done.

Line 315: “research area” – clarify

Rephrased to clarify we refer to the hut housing the experiment (l.505).

Line 373 (and throughout the manuscript): Authors have used some time “experimenter” and some time the author initials – need for consistency (i.e. always or never initials of individual experimenters)

All usage of initials has been switched to a neutral term.

Line 387 (and above): Ad libitum needs to be written in italics (as it comes from Latin)

Done.

Line 387: Add further citation to Altman (1974)

Done.

Discussion / Conclusions

Line 570: Which studies? Add citation(s)

Sentence has been deleted.

Lines 570-577: Reword these sentences – authors should not oversell their study findings

These sentences have been completely replaced so as not to oversell our study results or unfairly contrast our study to others (ll.784ff). We believe the resultant paragraph is much stronger than the original.

Line 609: Replace “came to understand” with “understood”

Done.

Line 612-613: Reword – “became good” is not formal writing style…

Fixed.

Reviewer 2 Report

I enjoyed the opportunity to review this paper. It appears to present a thorough introduction to the topic, and to provide interesting and well analysed data.

I just have a couple of questions I would like to ask.

  1. I would like more comment on the presence of the researcher during the experiments. Do you think that the presence increased or decreased the likelihood that the monkeys entered the hut and participated in the experiment? Should this presence be seen as a limitation?
  2. Do macaques communicate with each other? Would this communication have influenced participation and cooperation?
  3. Could the aggression seen ever have been a flow on from aggression displayed out in the enclosure and not be related to the task?

Thanks

Author Response

Reply to Reviewer 2

I enjoyed the opportunity to review this paper. It appears to present a thorough introduction to the topic, and to provide interesting and well analysed data.

Thank you for taking the time to go through our study and provide the insightful questions! We’ll address them in turn below.

I just have a couple of questions I would like to ask.

I would like more comment on the presence of the researcher during the experiments. Do you think that the presence increased or decreased the likelihood that the monkeys entered the hut and participated in the experiment? Should this presence be seen as a limitation?

Because the monkeys were well habituated to the presence of humans, we did not anticipate that the presence of the experimenter would have much effect on their subsequent behavior. That being said, the experimenter was always careful to step away from the cooperation device and move to the far wall after loading it so as to be as out of the way as possible while still being able to accurately record of the action. We have further clarified the role of the researcher and the habituated state of the monkeys in our methods (ll.439ff).

Do macaques communicate with each other? Would this communication have influenced participation and cooperation?

Japanese macaques are extremely vocal animals, and it is certainly not outside the realm of possibility that they could be using it for active coordination or recruitment. This question was unfortunately outside of the scope of our study, but we have added a couple sentences to the “Future Directions” section of our paper indicating this as a possible avenue of study for future research (ll.1156ff).

Could the aggression seen ever have been a flow on from aggression displayed out in the enclosure and not be related to the task?

Yes, it absolutely could have (and probably was in some cases). But the fact remains that we have demonstrated statistically that there is a connection between reward division outcome and the expression of aggression. Though the aggression we observed was probably in some cases spillover from outside of the experiment, it would not be likely to have occurred systematically in the pattern that we observed. Our results still then support our claims regardless of whether some aggression was due to factors beyond our study.

Round 2

Reviewer 1 Report

The authors have done well to address my comments from their previous submission. I believe the manuscript has been substantially improved. I am therefore glad to recommend this amended manuscript for publication in Animals.

However, I still have some reservations about stating that the study group is a "semi-free population of provisioned Japanese macaques". At least within the Abstract the study subjects should be defined as they are - i.e. “captive”. The authors can then use the Methods to describe the housing conditions of the macaques with full details and so ensure that readers do not misinterpret their living conditions, as well as use the term “semi-free” throughout the manuscript by explaining that this is used as already done by other authors who conducted studies under similar conditions.

Author Response

The authors have done well to address my comments from their previous submission. I believe the manuscript has been substantially improved. I am therefore glad to recommend this amended manuscript for publication in Animals.

We would like to thank the reviewer once more for their many helpful suggestions throughout the review process. We as well believe that the manuscript has been greatly improved after taking them into account. We have addressed your next set of concerns as indicated below.

However, I still have some reservations about stating that the study group is a "semi-free population of provisioned Japanese macaques". At least within the Abstract the study subjects should be defined as they are - i.e. “captive”. The authors can then use the Methods to describe the housing conditions of the macaques with full details and so ensure that readers do not misinterpret their living conditions, as well as use the term “semi-free” throughout the manuscript by explaining that this is used as already done by other authors who conducted studies under similar conditions.

We have changed the wording of the abstract to "a captive population of Japanese macaques" (l.32) and at our first mention of the term semi-free (in the introduction), we now draw attention to the size and conditions of the enclosure and cite other studies that have used the term in a similar manner (ll.114ff) in addition to the more thorough description included in Study Subjects section of our Methods.